# Review on Pediatric Malignant Focal Liver Lesions with Imaging Evaluation: Part I

**DOI:** 10.3390/diagnostics13233568

**Published:** 2023-11-29

**Authors:** Yi Dong, Andrius Cekuolis, Dagmar Schreiber-Dietrich, Rasa Augustiniene, Simone Schwarz, Kathleen Möller, Nasenien Nourkami-Tutdibi, Sheng Chen, Jia-Ying Cao, Yun-Lin Huang, Ying Wang, Heike Taut, Lara Grevelding, Christoph F. Dietrich

**Affiliations:** 1Department of Ultrasound, Xinhua Hospital, Shanghai Jiao Tong University School of Medicine, Shanghai 200092, China; drdaisydong@hotmail.com (Y.D.); 21211210033@m.fudan.edu.cn (S.C.); califfa@126.com (J.-Y.C.); 22111210067@m.fudan.edu.cn (Y.-L.H.); yunxi2009@126.com (Y.W.); 2Ultrasound Section, Department of Pediatric Radiology, Radiology and Nuclear Medicine Centre, Vilnius University Hospital Santaros Klinikos, 08661 Vilnius, Lithuania; andrius.cekuolis@gmail.com (A.C.); augustiniene.rasa@gmail.com (R.A.); 3Localinomed and Pediatric Department, Kliniken Salem, 3013 Bern, Switzerland; dietrich.dagmar@googlemail.com; 4Department of Neonatology and Pediatric Intensive Care Medicine, Sana Kliniken Duisburg GmbH, 47055 Duisburg, Germany; simone.schwarz@sana.de; 5Medical Department I/Gastroenterology, SANA Hospital Lichtenberg, 10365 Berlin, Germany; k.moeller@live.de; 6Saarland University Medical Center, Hospital of General Pediatrics and Neonatology, 66421 Homburg, Germany; nasenien.nourkami@uks.eu; 7Children’s Hospital, Universitätsklinikum Dresden, Technische Universität Dresden, 01069 Dresden, Germany; heike.taut@uniklinikum-dresden.de; 8Department of Pediatrics, Division of Pneumology, Allergology, Infectious Diseases and Gastroenterology, University Hospital Frankfurt, Goethe University, 60323 Frankfurt, Germany; 9Department Allgemeine Innere Medizin (DAIM), Kliniken Hirslanden Beau Site, Salem und Permanence, 3013 Bern, Switzerland

**Keywords:** pediatric, malignant, focal liver lesions (FLLs), diagnosis, imaging, contrast enhanced ultrasound (CEUS)

## Abstract

Malignant focal liver lesions (FLLs) are commonly reported in adults but rarely seen in the pediatric population. Due to the rarity, the understanding of these diseases is still very limited. In children, most malignant FLLs are congenital. It is very important to choose appropriate imaging examination concerning various factors. This paper will outline common pediatric malignant FLLs, including hepatoblastoma, hepatocellular carcinoma, and cholangiocarcinoma and discuss them against the background of the latest knowledge on comparable/similar tumors in adults. Medical imaging features are of vital importance for the non-invasive diagnosis and follow-up of treatment of FLLs in pediatric patients. The use of CEUS in pediatric patients for characterizing those FLLs that remain indeterminate on conventional B mode ultrasounds may be an effective option in the future and has great potential to be integrated into imaging algorithms without the risk of exposure to ionizing radiation.

## 1. Introduction

Hepatic pathologic masses in pediatric patients include primary neoplasms and metastatic lesions from distant malignancies, inflammatory masses, and cysts [1]. Approximately two-thirds of all focal liver lesions in pediatric patients are malignant and account for the third most common solid intra-abdominal malignancies in pediatric patients [2,3]. Pediatric liver tumors are systematized and standardized in the International Consensus Classification in Los Angeles 2011 [4].

Among all primary malignant hepatic tumors in pediatric patients, hepatocellular carcinoma (HCC), typically affects older children and adolescents [5]. In addition to hepatocellular malignant tumors, liver epithelial malignant tumors also comprise biliary malignant tumors, including cholangiocarcinoma (CCA) and combined HCC-CCA. The evaluation of pediatric liver tumors is largely driven by clinical manifestations, age at diagnosis, presence of any medical comorbidities, serum alpha-fetoprotein (AFP) level, and imaging characteristics [6]. Imaging modalities are used to determine the organ of origin, the character of the lesion, and the extent of the lesion in pediatric patients [7]. Therefore, a thorough understanding of pediatric liver tumors and associated imaging features is required to achieve an accurate diagnosis while evaluating all potential differential diagnosis. However, sedation and the risks of radiation exposure should also be taken into consideration when analyzing imaging modalities in children. Based on the children’s age and clinical conditions, differential diagnostic strategies should be tailored [8]. Ultrasound is used as the first line imaging method to detect the lesion and to provide a preoperative diagnosis and differential diagnosis [9]. Compared to computed tomography (CT) and/or magnetic resonance imaging (MRI), ultrasound is distinguished by certain advantages and disadvantages.

Overall, challenges exist in the detection and characterization of focal liver lesions (FLLs) non-invasively among the pediatric population and, depending on various factors, to the selection of appropriate imaging modalities. First, modern imaging technologies with higher resolution are required in the pediatric population due to smaller anatomic structures. Second, sedation or anesthesia might be required for some children who are unable to cooperate or control their breathing during an imaging examination due to their age and/or overall condition. Third, since children are more sensitive to radiation exposure, the use of imaging methods with potential radiation exposure should be minimized [10].

The aim of the current review is to discuss pediatric malignant FLLs with an emphasis on imaging characteristics and the benefits of each imaging modality on the background of current knowledge and treatment strategies of FLLs.

## 2. Imaging Evaluation on Pediatric Malignant Focal Liver Lesions

Advances in imaging techniques have significantly improved pediatric liver tumor diagnostic efficiency. Ultrasound, CT, and MRI are commonly used in clinical practice.

### 2.1. Ultrasound

A conventional ultrasound scan is a convenient, real-time, non-invasive, and radiation-free modality and, therefore, it is particularly utilized for the diagnosis, screening, and clinical follow-up of FLLs in pediatric patients [11]. It can instantly determine the number, size, location, and appearance of FLLs and helps in narrowing the differential diagnosis [12]. Liver parenchyma and portal venous involvement could also be evaluated. Color Doppler imaging can deliver information on the potential vascular nature of the tumor [8]. More recently, elastography is applied as an additional functional ultrasound technique offering the quantitative evaluation of tissue stiffness. Elastographic evaluation reflects tissue architecture and composition, perfusion, injury, and edema [13,14]. Ultrasound elastography is approved by the Food and Drug Administration (FDA) for the evaluation of solid abdominal organs in children. At the same time, even though the rate of major complications is low, ultrasound-guided biopsy procedures have reported a high rate of minor bleeding in pediatric patients not requiring further intervention [15].

Nevertheless, inherent limitations and the frequently inconclusive nature of conventional ultrasound may require extra cross-sectional imaging such as CT or MRI to better characterize FLLs and improve diagnostic accuracy, with the requirement of the administration of contrast inevitably in CT and more frequently in MRI [16]. Both imaging techniques are associated with risks for pediatric patients, especially pediatric oncology patients, including exposure to ionizing radiation by CT, the use of iodine- or gadolinium-based contrast agents, and the potential need for sedation or anesthesia and antihistamines in case of allergic reactions to contrast agents [11]. Unlike CT or MRI contrast agents, ultrasound microbubble contrast agents are exhaled, eliminating concern for the need for sedation, radiation exposure, and related renal injury and are therefore particularly well-suited for the pediatric population [17]. More recently, liver applications of contrast enhanced ultrasound (CEUS) are more frequently reported in pediatric patients and CEUS is increasingly integrated into imaging algorithms for the detection and characterization of FLLs [18,19,20,21,22,23]. CEUS allows for the real-time imaging of dynamic perfusion and quantitative analysis of wash-in and wash-out contrast kinetics [13]. Due to different vascularization patterns, the typical CEUS patterns of malignant lesions include fast wash-in during the arterial phase and rapid wash-out during the portal venous and late phases [24]. Currently, three available ultrasound contrast agents have been used in children for liver imaging, including SonoVue/Lumason, Optison, and Definity.

Focal liver lesions are one of the FDA (U.S. Food and Drug Administration) approved applications of CEUS with the utilization of the UCA Lumason in children, along with echocardiography and intravesical application [25]. CEUS in children is relatively safe, as an example in 57 pediatric-only studies involving 4906 intravenous CEUS applications, the rate of serious adverse events was 0.22% and the rate of nonserious adverse events was 1.2% in children [25]. The recommended dosage for the intravenous administration of SonoVue/Lumason is either age- and/or weight-dependent [17]. The FDA recommends 0.03 mL/kg body weight per application for Lumason, with a maximum of 2 applications per exam. Other recommendations are 0.1 mL/year of age for children under 6 years of age [26] or 0.6 mL up to 6 years of age, 1.2 mL up to 12 years of age, and a maximum of 2.4 mL over 12 years of age [27]. CEUS is particularly useful in children with unclear liver lesions in the setting of preexisting liver or systemic disease to prove or disprove the benignity of the lesions. In a study conducted exclusively on children, a specificity of 98% and negative predictive value of 100% were described [28].

### 2.2. Computed Tomography (CT)

Advancements in technology and the availability of multi-detector computed tomography (MDCT) have proven to be essential diagnostic imaging modalities, with an increasing range of examinations now performed on the pediatric population [29]. The ability of MDCT to acquire the image data of pediatric FLLs and surrounding structures with a fast scanning time and high spatial resolution allows for the accurate anatomical localization and visualization of FLLs, with an augmentation of diagnostic accuracy [30]. Moreover, multiphase contrast enhanced computed tomography (CECT) allows for the further characterization of FLLs to improve diagnostic specificity [10]. The use of multi-slice helical computed tomography (MSCT) allows for shorter imaging time enabling the reduction of the need for sedation, including its risks [31]. However, the risks of an increased radiation burden and associated potential hazards in children cannot be ignored. A good understanding of the relationship between image quality and radiation dose is required to optimize scanning parameters [29].

### 2.3. Magnetic Resonance Imaging (MRI)

Due to its superior soft tissue contrast resolution, MRI is preferable for the evaluation of FLLs with excellent multi-planar spatial resolution and functional assessment while avoiding the exposure to the ionizing radiation inherent to CT and nuclear medicine imaging techniques [32,33]. Pediatric tumor characterization on an MRI relies on the determination of the T2 signal and enhancement features, which is helpful for the diagnosis of FLLs [34]. Specific MRI contrast agents for the liver are helpful for the diagnosis and characterization of FLLs, being a non-invasive and relatively rapid to perform imaging approach [10]. It has been reported that with the application of liver specific extracellular contrast agents, it could achieve the better characterization and localization of lesions, particularly with respect to the biliary system [33]. The major limitations of MRI comprise the requirement for sedation in pediatric patients and longer imaging duration, urging the need for MRI improvements, especially with regard to faster sequences [10].

### 2.4. PRE-Treatment EXTent of Tumor (PRETEXT) System

The PRE-Treatment EXTent of tumor (PRETEXT) system is a radiologic staging system for the evaluation of primary hepatic malignancies in pediatric patients and is usually used in the risk stratification for hepatoblastoma, pediatric HCC across the world, and consists of the PRETEXT group and the annotation factors [7]. The PRETEXT group (I, II, III, IV) describes the extent of the tumor within the liver based on determining the number of contiguous tumor-free liver sections, which has been confirmed to be a powerful predictor of overall survival in pediatric patients with HCC and hepatoblastoma [35,36,37,38]. The PRETEXT group can be determined by calculating the number of contiguous sections that would have to be resected to completely remove the tumor. The annotation factors help to describe associated features, including hepatic venous/inferior vena cava involvement (V), portal venous involvement (P), extrahepatic disease contiguous with the main liver tumor (E), multifocality (F), tumor rupture (R), caudate (C), lymph node metastases (N), and distant metastases (M), which are used to help risk-stratify patients [7].

## 3. Hepatoblastoma

### 3.1. Epidemiology, Clinical Features, and Pathological Features

Among all primary liver tumors in pediatric patients, hepatoblastoma is the most common one accounting for almost two-thirds of all pediatric malignant liver tumors [39]. Hepatoblastoma can occur in children of any age, predominantly in children between 6 months and 3 years of age [40]. Hepatoblastoma may also occur in older children up to 15 years of age [3]. A male-to-female predominance of 1.6 to 1.0 has been reported [2,41]. The incidence of hepatoblastoma varies from 0.5 to 2 cases per million children per year [42]. Various genetic diseases have been found to be associated with hepatoblastoma, such as familial adenomatous polyposis, Beckwith–Wiedemann syndrome, and trisomy 18 (Edwards syndrome) [43,44,45]. In addition, prematurity and very low birth weight are known to increase the risk of developing hepatoblastoma [40].

Hepatoblastoma can remain asymptomatic for months, with painless swelling probably occasionally occurring in the upper right abdominal area in the early stages [46]. In general, children with hepatoblastoma most often present with an enlarged, distended abdomen accompanied by nonspecific clinical symptoms such as nausea, vomiting, weight loss, and increasing overall weakness [47]. In advanced stages, pediatric patients begin to present with more symptoms. Pathological fractures, obstructive jaundice, metastasis, and spontaneous tumor rupture with extensive bleeding are rarely observed [48,49,50,51]. In addition, hepatoblastoma may secrete human chorionic gonadotropin, resulting in precocious puberty in boys [52]. The most useful laboratory test for hepatoblastoma is alpha-fetoprotein (AFP), showing abnormal elevation in 80–90% of patients, and therefore serving as a sustainable biomarker for diagnosis, monitoring, and disease follow-up [1]. Hepatoblastoma with low levels of AFP is considered aggressive and has abysmal prognosis or is associated with the small cell undifferentiated (SCUD) subtype [53].

Hepatoblastoma, derived from degenerated hepatoblasts, generally appears as a well-circumscribed, encapsulated, solitary, large solid liver mass with lobulated contours, while a small part may present as multiple lesions [54]. Hepatoblastomas are histologically classified broadly into epithelial type, mixed epithelial type, and mesenchymal type tumors. Depending on different degrees of differentiation, hepatoblastoma epithelial cells can be divided into embryonic cells being poorly differentiated and fetal cells being well-differentiated, both able to be present at the same time. In addition, mesenchymal stroma includes collagen, osteoid, skeletal muscle cells, and cartilage [8]. Small-cell undifferentiated hepatoblastoma is a unique subtype, which may not be associated with raised AFP levels and is considered prognostically unfavorable [55].

### 3.2. Imaging Features

**Ultrasound:** On conventional B-mode ultrasound, hepatoblastoma can present mostly as a solitary mass, a dominant mass with satellite lesions, multiple nodules throughout the liver, or rarely a diffuse infiltrative mass involving the whole liver (Figure 1) [56]. Most hepatoblastoma lesions present as hyperechoic lesions in comparison to the surrounding liver parenchyma [8]. Epithelial hepatoblastoma typically demonstrates a relatively more homogeneous appearance, while mixed epithelial and mesenchymal hepatoblastomas present as heterogeneous tumors due to osteoid, cartilaginous, and fibrous components [17]. Coarse calcifications are present in 20–50% of all cases and appear as punctiform or linear hyperechoic foci with posterior shadowing. Areas of necrosis or hemorrhage within the tumor are commonly seen and present with an anechoic appearance [56]. Zhuang et al. [57] evaluated the use of grey-scale ultrasound for differentiation of hepatoblastoma from HCC in the pediatric population and showed that septa (25/30, 83.3% vs. 2/12, 16.7%, *p* < 0.001) and liquefaction (17/30, 56.7% vs. 3/12, 25.0%, *p* = 0.02) were more frequently found in hepatoblastoma than in HCC.

Color Doppler imaging can be used to detect the presence of high velocities within tumor tissue and the invasion of the hepatic and portal veins, strongly supportive for the diagnosis of malignant tumors [58]. Pan et al. [59] studied the ultrasound features of pediatric patients for the ability to distinguish infantile hepatic hemangioma (IHH) from hepatoblastoma and reported that the arterial flow that could be detected in hepatoblastoma was with relatively higher resistance indices (RI) (>0.7 [15/20, 75.0% vs. 1/8, 12.5%, *p* = 0.004]), while IHH was characterized by arterial flow with RI lower than (<0.7) and/or venous flow. Additionally, tumor thrombi in the portal or hepatic veins may also be observed [3]. However, the assessment of the portal and hepatic veins by Doppler ultrasound may be a challenge when tumors are large, and vessels are compressed. In this case, CT and/or MRI may additionally be performed to confirm the final diagnosis [56]. Although the ultrasound features of hepatoblastoma may overlap with other pediatric FLLs, it is nonetheless valuable for a preliminary examination.

**CEUS:** Like other liver malignancies, during the arterial phase, hepatoblastoma can show early peripheral hyperenhancement. During the late portal venous phase, it shows marked washout [17,60]. Wang et al. [61] analyzed the CEUS features of FLLs in pediatric patients. Their results showed CEUS features of hyperenhancement during the arterial phase and early washout (≤45 s), and showed a sensitivity of 90.7% and specificity of 93.6% to predict hepatoblastoma in pediatric patients <5 years. The study also developed a pediatric liver CEUS criterion based on the diagnostic features of FLLs on CEUS, demonstrating that CEUS is helpful in differentiating pediatric malignancies from benign FLLs, particularly hepatoblastoma from hepatic hemangioma. Apart from this, CEUS can accurately identify tumor thrombi as enhancing filling defects within invaded vessels in case of the portal or hepatic veins being invaded by the tumor tissue of hepatoblastomas [17]. However, CEUS is limited in the utility for the tumor staging of primary liver malignancy, which is of vital importance to select the appropriate treatment for hepatoblastoma [62].

**CT:** On unenhanced CT, hepatoblastoma appears extremely variable, due to the histologic composition of the tumor. It typically presents as a well-defined, slightly hypodense mass compared to the adjacent liver parenchyma. Some hepatoblastomas, such as epithelial hepatoblastomas, appear more homogeneous, while the mixed type hepatoblastomas appear to be mostly heterogeneous. Calcifications can be observed in more than half of the hepatic lesions and are usually small, fine lesions found in epithelial type hepatoblastomas and are more gross and extensive in mixed type hepatoblastomas.

Following the injection of contrast agents, hepatoblastoma generally shows a heterogeneous pattern of enhancement, sometimes revealing a peripheral hyperdense rim in the early arterial phase and an iso- or hypodense rim on delayed images. Approximately half of hepatoblastoma lesions appear lobulated or septate, especially on contrast enhanced images. Baheti et al. [63] studied 34 pediatric patients with hepatoblastoma who underwent CECT examination and revealed that irregular tumor margin was the only imaging feature significantly associated with aggressive tumor behavior. Importantly, the invasion of the portal vein and its subsequent thrombi must be evaluated in all suspected cases of hepatoblastoma which may spread along the inferior vena cava to the lumen of the right atrium. Regarding the latter, CT scans are a convenient and indicated choice to additionally detect pulmonary metastases and complete tumor staging. Kim et al. [64] evaluated lung metastasis in pediatric patients with hepatoblastoma based on a staging chest CT, and found that 28% of patients with hepatoblastoma had lung metastasis with an overall accuracy of 96.8% of staging chest CT. Multifocality and male sex were predictors of lung metastasis.

**MRI:** MRI provides multiplanar images of primary liver tumors with excellent depiction of vascular anatomy. Epithelial hepatoblastoma typically shows as homogeneous and slightly hypointense on T1-weighted images and hyperintense on T2-weighted images. Mixed hepatoblastoma demonstrates more heterogeneous signal intensity features due to varying amounts of components as noted [3]. Fibrotic septa appear on MRI as hypointense bands on both T1- and T2-weighted images and show hyperenhancement after intravenous administration of gadolinium contrast agents. Areas of hemorrhage inside of the tumor will show as hyperintense on T1-weighted images and central necrosis may appear hyperintense on T2-weighted images. Calcifications could be detected in 50% of cases, but are not well seen on MRI [8]. Vascular invasion could be well demonstrated by gradient-echo imaging or contrast enhanced magnetic resonance angiography (MRA), being helpful in preoperative surgery planning to detect normal anatomic or possible vascular variations important to set surgical margins [56]. Diffusion-weighted imaging (DWI) can be a better tool for the assessment of the satellite lesions of hepatoblastoma which are usually missed on CECT. Sharma et al. [65] reported two cases of hepatoblastoma where more satellite lesions were detected on DWI compared to multiphase CECT (8 lesions vs. 1 lesion; 6 lesions vs. no lesion). The latter might be crucial for proper staging, treatment planning, and patient outcome.

On contrast enhanced magnetic resonance imaging (CEMRI), hepatoblastoma typically reveals a heterogeneous enhancement pattern, which is usually hypointense compared to adjacent liver parenchyma during all phases of post-contrast imaging [33]. Mayers et al. [66] reported one variant of hepatoblastoma that presented with an area of central enhancement during the hepatocyte phase of imaging, which might be related to its teratoid features on pathology.

**Other imaging techniques:** A benefit of positron emission tomography (PET) is the ability of whole-body imaging, which may be useful in realizing or excluding possible bone metastases. Zhang et al. [67] reported one case of a 4-year-old girl with a relapse of hepatoblastoma. However, CT, abdominal ultrasound, and MRI scans could not determine the site of tumor recurrence. Increased activity in the region of the left scapula and adjacent soft tissue could be shown by F-FDG PET/CT. Meanwhile, increased activity and a noticeable progressed lesion in the same place could be shown by F-DOPA PET/CT and be proved by histopathological examination results. However, there is lack of proof whether FDG-PET/CT is suitable to be used as a first line imaging method for an initial diagnosis of possible hepatoblastoma.

### 3.3. Management and Prognosis

Surgical resection with pre- and/or post-operative chemotherapy forms the mainstay of therapy for hepatoblastoma, with some cases of hepatoblastoma remaining unresectable due to large size, multifocality, and/or location close to major vascular structures [68]. Neoadjuvant chemotherapy including platinum compounds renders the tumor amenable to resection. Upfront neoadjuvant chemotherapy generally leads to a significant tumor volume reduction in many pediatric solid tumors and results in a significant improvement of prognosis and patient survival [69]. Despite aggressive neoadjuvant chemotherapy, hepatoblastoma cannot be completely resected in approximately 10% of PRETEXT IV children and orthotopic liver transplant must be considered in those cases [70]. Due to excellent multidisciplinary approaches and therapy regimens, survival following liver transplantation is excellent with survival rates of more than 85% [71].

The 5- and 10-year overall reported survival in pediatric patients with hepatoblastoma were 81.5% and 81.0%, respectively [72]. Ke et al. [73] respectively analyzed 311 pediatric patients with hepatoblastoma who underwent surgical resection and multivariate analysis and suggested that age, histological pattern, microvascular invasion, multifocality, distant metastasis, and macrovascular invasion are independent prognostic factors in hepatoblastoma. About 12% of pediatric patients with hepatoblastoma achieving complete remission are likely to relapse. Therefore, regular follow-ups are of vital importance to monitor long-term treatment effects [74].

## 4. Hepatocellular Carcinoma (HCC)

### 4.1. Epidemiology, Clinical Features, and Pathological Features

Pediatric HCC is the 2nd most common pediatric malignant liver tumor after hepatoblastoma, accounting for approximately 1/5 to 1/3 of all liver cancers in pediatric patients [5]. Unlike hepatoblastoma, pediatric HCC typically affects adolescent children aged 10–14 years with a slight male predominance [8]. The overall incidence of HCC is estimated to be 0.41 (0.24–0.65) per million per year in pediatric patients, depending on the country of origin and prevalence of Hepatitis-B virus (HBV) infections [5,39,75]. Although HCC in adulthood occurs primarily in the background of liver cirrhosis, approximately 70% of pediatric HCCs occur without any history of cirrhotic liver disease [76]. According to current data and reports, there is no known geographic and/or demographic endemic for viral hepatitis [77]. In addition to hepatitis B and hepatitis C also associated with HCC in adults, various genetic diseases are known to constitute predisposing conditions for the development of pediatric HCC, such as glycogen storage disease, tyrosinemia, Alagille syndrome, alpha1-antitrypsin deficiency, and progressive intrahepatic familial cholestasis type 2 [2,78,79]. Other predisposing risk factors related to pediatric HCC, include biliary atresia, parenteral nutrition-associated liver disease, autoimmune hepatitis, primary sclerosing cholangitis, and hemochromatosis [3,80].

The common symptoms of pediatric HCC include abdominal mass, abdominal pain, weight loss, and other nonspecific symptoms. In advanced cases, children often have cachexia, jaundice, and gastroesophageal reflux. Symptoms and signs of portal hypertension and decompensated end-stage liver disease, such as variceal bleed, encephalopathy, spider naevi, ascites, and clubbing, occur in children with chronic liver disease and/or liver cirrhosis. Splenomegaly and ascites are associated with the degree of portal hypertension [5,75]. Serum AFP levels increase in approximately 50% to 70% of patients, and typically exceed up to 200 ng/mL [81].

Three pathologic types of HCC could be seen in pediatric patients. Classic HCC is the most common, resembling adult type HCC. Fibrolamellar HCC is a distinctive neoplasm arising in non-cirrhotic liver as large, solitary, and circumscribed mass. Fibrolamellar variants are composed of large cells and copious eosinophilic cytoplasm in a lamellated stroma [75]. This form occurs mostly in adolescents and young adults, accounting for about 24% of all HCC cases, with a far better prognosis [76]. The incidence is independent of any known liver disease [4]. “Hepatocellular neoplasm not otherwise specified” (HCN-NOS) is a rare, transitional histological entity which shows both hepatoblastoma and HCC features [4,82,83]. Macroscopically, pediatric HCC can occur as solitary, multinodular, and rarely diffusely infiltrative growing tumors [8]. Microscopically, the morphological appearance of HCC in pediatric patients including neoplastic cells varies widely from well-differentiated cells to poorly differentiated cells [4]. Histologically, the most common pattern is trabecular. Other patterns have also been reported including pseudo-glandular, clear-cell, acinar, steatosis, compact, and scirrhous [84,85]. Various grading systems, including Edmondson–Steiner grade that divides tumors into low grades (grade I and II) and high grades (grade III and IV), have been described [86]. According to a proposed WHO (2019) grading system for hepatocellular carcinoma, HCC is graded as well/moderately and poorly differentiated [87]. Microvascular invasion and poorly differentiated tumors have proven to be independent risk factors for poor outcome [32].

### 4.2. Imaging Features

**Ultrasound:** Pediatric HCC typically presents with a heterogeneous, predominantly hyperechoic structure with variable size with increased vascularity [75]. Anechoic areas in tumors due to hemorrhage or necrosis can be observed. A thin hypoechoic halo can also be detected in tumors with capsules [88]. In contrast, smaller HCC lesions may tend to present a hypoechoic or hyperechoic appearance. Infiltrative HCC may appear as a diffuse abnormality of liver echogenicity. High-velocity arterial flow could be detected by color Doppler imaging. Ultrasound scans are helpful in detecting HCC tumor infiltration of the portal and/or hepatic veins in pediatric patients [8].

Prior studies have shown that pediatric HCC tends to be larger in size than adult HCC at initial presentation. The latter might be due to a higher basic cellular liver growth rate in infants and children [89]. In addition, pediatric HCC occurs without any history of liver disease, therefore standardized surveillance guidelines for those with predispositions, as in adults, are lacking for children. Rees et al. [90] evaluated the imaging findings of nonfibrolamellar pediatric HCC and their associations with the presence of predisposing factors. The results showed that patients with versus those without predispositions exhibited smaller tumor size (6.0 cm vs. 11.9–12.9 cm, *p* < 0.05), less tumor infiltration into veins (0% vs. 36–41%, *p* < 0.05), and lower incidence of PRETEXT stage IV (18% vs. 50–55%, *p* < 0.05), likely due to surveillance imaging. The study supports the role of routine surveillance imaging in pediatric HCC with predispositions to facilitate early tumor detection.

**CEUS:** The use of CEUS in adults for the diagnosis and differential diagnosis of HCC has been well described [91]. Pediatric HCC likely appears similar to HCC in adults and potentially shows hyperenhancement during the arterial phase and subtle late washout. Although CEUS may be considered for problem-solving, CT and/or MRI are required for the staging of HCC [92] [Figure 2].

**CT:** Pediatric HCC on CT may show variable appearances and nonspecific features such as slightly hypodense or isodense, solitary or multiple, heterogeneous or homogeneous, and ill- or well- defined features [3]. Due to the presence of components of hemorrhage, necrosis, fat, and calcification, mosaic appearance is more typical in larger lesions [32]. During the arterial phase of CECT, the typical characteristics of pediatric HCC are early hyperenhancement. In the portal venous phase, the lesion shows rapid washout and is usually inconspicuous on delayed scans [2]. The lesions sometimes may not demonstrate washout during the portal venous phase. The tumor capsule may be observed as rim-like hypointense and may enhance in the delayed phase [8]. In addition, the invasion of important vessels, such as the portal veins, hepatic arteries, hepatic veins, and inferior vena cava could be observed. Arias et al. [93] described the imaging features of pediatric HCC in 15 children and arterial phase hyperenhancement could be detected in 83% of cases, washout in 86%, tumor-in-vein in 33%, and a capsule in 50%. The study revealed that as reflected by high PRETEXT staging and commonly invasion to the portal vein and caudate involvement, pediatric HCCs are highly heterogeneous malignant tumors, which should be taken into consideration when determining the resection margins of these tumors at presentation. In addition to the further differentiation of the lesions by CT, chest CT is recommended to rule out pulmonary and/or thoracal bone metastases [94].

**MRI:** On MRI, HCC in the pediatric population usually likewise shows as variable but predominantly slightly hypointense on T1-weighted images and mildly hyperintense on T2-weighted images, and heterogeneous signal intensity is often seen in larger lesions [3]. After intravenous administration of contrast agents, pediatric HCC presents with a similar pattern on CT with early avid enhancement in the arterial phase as well as prompt washout and reduced signal intensity in the portal venous phase, with significant implications for clinical staging and surgical intervention [95]. On T1- and T2-weighted images, the fibrous capsule may show as hypointense. During the delayed phase, it demonstrates enhancement [96]. On gadolinium-enhanced MRI, vascular invasion may be observed as a lack of a signal void on spin-echo images, an arterial hyperenhanced mass, and a delayed filling defect [97]. Both DWI and Gadolinium ethoxybenzyl diethlenetriamine pentaacetic acid (Gd-EOB-DTPA)-enhanced MRI with the hepatobiliary phase are helpful in the detection, characterization, and diagnostic yield of pediatric HCC [32,98]. Most HCC lesions present as hypointense on the hepatobiliary phase.

The Liver Imaging Reporting and Data System (LI-RADS) proposed by the American College of Radiology (ACR) provides standardized criteria for reporting CT and MRI findings in adults at risk of developing HCC but not in the pediatric population [99]. Standardized image interpretation which can support the diagnosis of suspected hepatocellular neoplasms will be needed. Khanna et al. [100] evaluated the performance of LI-RADS version 2018 for the diagnosis of 58 cases of pediatric HCC and showed that LI-RADS version 2018 had moderate sensitivity (85–88%) but low specificity (54–70%). LI-RADS major criteria used for the diagnosis of adult HCC at risk are seen in only a subset of pediatric HCC and are often encountered in benign hepatic tumors in children. However, two well-described MRI features of diagnostic value should be recalled other than the LI-RADS criteria. On the one hand, it has been demonstrated that small and hypovascular HCCs are characterized by double hypointensity in the portal/venous and hepatobiliary phases that could be considered a magnetic resonance pattern highly suggestive of hypovascular hepatocellular carcinoma. On the other hand, abbreviated MRI (AMRI), corresponding to the acquisition of a limited number of sequences including T1-weighted, T2-weighted, and DWI sequences, with or without contrast administration, has been suggested as an accurate screening imaging tool by recent studies.

**Other imaging techniques:** Areas of high metabolism may aid in the detection of extrahepatic locations that may be missed by other imaging techniques. Pessanha et al. [101] investigated the effects of preoperative FDG-PET on the prediction of HCC recurrence after liver transplantation. Their results demonstrated that PET-positive status was significant and an independent risk factor for recurrence. Preoperative FDG-PET images offer potential information to predict the risk of HCC recurrence following liver transplantation. The latter may also be useful in pediatric patients.

**Imaging features of fibrolamellar HCC:** Fibrolamellar HCC typically demonstrates as a solitary well-defined mass with a heterogeneous but predominantly iso- or hyperechoic appearance and a hyperechoic central scar on ultrasound [102]. Dong et al. [103] evaluated the CEUS features of histologically proven fibrolamellar HCC in 16 patients ranging from 16 to 35 years of age in comparison to benign focal nodular hyperplasia (FNH). Although fibrolamellar HCC and FNH showed similar non-enhanced central scars, fibrolamellar HCC presented peripheral hyperenhancement in the arterial phase as well as early washout and hypoenhancement in the portal venous and late phase as a sign of malignancy while all FNH showed hyperenhancement (*p* < 0.01). CEUS could reliably diagnose fibrolamellar HCC as a malignant FLL and show differentiation between fibrolamellar HCC and FNH lesions.

Fibrolamellar HCC exhibits similar features to classic HCC on unenhanced CT (hypodense) or MRI and CECT or CEMRI [104]. The central scar is a distinctive feature of fibrolamellar HCC that typically shows as hypointense on T1- and T2-weighted images, which is different form FNH, in which the central scar is hyperintense on T2-weighted images and shows enhancement on the delayed phase, and helps to rule out or to prove the potential differential diagnosis [32]. The study of Arias et al. [93] revealed that in pediatric patients’ normal liver background, a central scar and normal AFP level may help differentiate fibrolamellar HCC from other types of HCC.

**Imaging features of HCN-NOS:** HCN-NOS is a new provisional entity that demonstrates complex morphologies and an admixture of histological patterns typical of both HCC and hepatoblastoma in the same tumor in some cases [105]. These tumors have an aggressive nature and are obviously associated with high serum AFP levels [105]. On imaging, HCN-NOS lesions, usually presenting with PRETEXT stage III/IV, tend to show large size and multifocality at initial presentation with aggressive features such as major vascular involvement, metastases, and extrahepatic extension [32].

### 4.3. Management and Prognosis

Treatment of pediatric HCC is challenging, including surgical tumor resection, systemic chemotherapy, targeted therapies, liver transplantation, and radiological interventions such as radiofrequency ablation (RFA) and transfemoral hepatic artery chemoembolization (TACE). Complete surgical resection is essential for curative treatment, but approximately more than 80% of pediatric patients present with unresectable HCC with advanced stage with vascular invasion, multifocal involvement, or extrahepatic metastasis [89,106,107,108]. Tumors with PRETEXT stage I/II are quite easily removed surgically, but PRETEXT stage III tumors require further liver transplantation and intensive care facilities. Due to substantial liver involvement, tumors with PRETEXT stage IV are considered as unresectable. Importantly, intraoperative ultrasound is helpful for the real-time determination of safe resection borderlines [2]. Pediatric HCC is reported to have a relatively higher response to chemotherapy based on the PLADO course (cisplatin and doxorubicin) as compared to HCC in adults (up to 50% vs. 30%) [77,109]. Schmid et al. [110] evaluated the experience of PLADO in combination with sorafenib in pediatric HCC patients, which may be a promising approach to improving survival. For pediatric patients with advanced, unresectable, and non-metastatic HCC, liver transplantation should be considered at the earliest possible opportunity, especially in the context of pre-existing chronic liver disease [111]. Adults with HCC who require transplant are evaluated according to the Milan criteria, which may not be applicable to HCC in pediatric patients, and individual decisions should be made [112]. Moreover, for those patients with chemotherapy-resistant liver tumors, TACE might be a palliative option for those that are awaiting transplantation or potential surgical resectability [107]. Kohorst et al. [111] described a multimodal treatment, comprising chemotherapy with PLADO, TACE, timely liver transplantation, and post-transplant therapy with sorafenib and sirolimus (mTOR inhibitor), that may aid in improving outcomes and prolonging survival rates in children with unresectable HCC.

Overall, due to the development of surgical techniques and medical imaging follow-up methods, the 5-year survival rate of pediatric patients with non-metastatic, resectable HCC might increase up to 70–80%, while it remains less than 20% in children with unresectable HCC [106,107]. Wang et al. [113] retrospectively compared overall survival of 65 HCC patients aged no more than 20 years and showed that, overall survival with resection, TACE, and supportive treatment was 38.0, 13.6, and 1.8 months, respectively, in moderate stage tumor disease. In addition, TACE offered a survival benefit in comparison to supportive treatment (7.1 vs. 2.3 months, *p* = 0.045). Previous research has shown that the PRETEXT stage, recurrence, vascular invasion, tumor size, and distant metastases are associated with the outcome of liver transplantation in pediatric patients with HCC [5]. According to the Surveillance, Epidemiology, and End Results (SEER) database, liver transplantation had better 5-year survival rates compared with resection (85% vs. 53%, hazard ratio, 0.05) [114]. It is reported that pediatric patients with HCC based on inherited disease have a potential survival advantage when transplantation is considered. Baumann et al. [115] evaluated 175 children who underwent liver transplantation for HCC and survival analyses showed the better long-term survival of pediatric patients with inherited liver disease compared with pediatric HCC patients without inherited liver disease (hazard ratio, 0.29) and adult HCC patients with inherited liver disease (hazard ratio, 0.27). Moreover, due to an earlier stage at diagnosis or more favorable opportunities for surgical resectablity, fibrolamellar HCC has a far more favorable 1-year and 5-year overall survival compared with non-fibrolamellar HCC [2].

## 5. Cholangiocarcinoma (CCA)

### 5.1. Epidemiology, Clinical Features, and Pathological Features

CCA is an adenocarcinoma from a bile duct that develops from the epithelial cells of the intrahepatic and extrahepatic bile ducts, primarily occurring in the adult population and exceedingly rare in children, accounting for less than 1% of all malignant FLLs in pediatric patients [116]. The overall incidence of CCA in those under the age of 20 years was reported to be 0.0036 per 100,000 per year. The median age upon diagnosis is 15 years (range 3–18 years), with boys having higher incidences than girls similar to adults [117]. The majority of pediatric patients with CCA were reported to have particular underlying risk factors, including congenital malformations of the biliary tree such as a choledochal cyst [118] and biliary atresia [119], primary sclerosing cholangitis [120,121], inflammatory bowel disease [122], primary immune deficiency [123], human immunodeficiency virus (HIV) infection [124], and post radiation therapy [125].

Abdominal pain, jaundice, pruritus, and elevated carbohydrate antigen 19-9 (CA19-9) are the most common clinical signs. Pediatric patients with intrahepatic cholangiocarcinoma (iCCA) may show fewer dominant strictures and obstructive symptoms than those with extrahepatic cholangiocarcinoma (eCCA) therefore making iCCA more difficult to be diagnosed clinically, especially in those patients with existing intrinsic liver disease, such as intrahepatic Caroli’s disease and progressive familial intrahepatic cholestasis [117].

The macroscopic type of iCCA can be subcategorized into the mass-forming type, the periductal infiltrating type, and the intraductal growing type [126]. Microscopically, CCA can be well to poorly differentiated adenocarcinoma and tumor cells are often arranged in tubules and glands with varied differentiation, typically dispersed in a fibrous stroma [32]. CCA is typically positive for CK7 and CK19, which might be helpful for differential diagnosis from HCC and hepatoblastoma. González et al. [127] presented the rare case of a solid-tubulocystic variant of iCCA in a 15-year-old girl, whose distinct pathologic features are essential to avoid confusion with neoplasms with similar appearances. In addition, combined HCC-CCA is an even rarer tumor that exhibits both hepatocytic and biliary differentiation and generally has a worse prognosis [128].

### 5.2. Imaging Features

**Ultrasound:** Ultrasound is generally used as an efficient method for routine screening and CCA surveillance of high-risk pediatric patients. Pediatric CCA usually shows as hypo- to isoechoic soft tissue on ultrasound. A round and dilatated common bile duct (CBD) associated with a dilatation of the intrahepatic bile ducts might be observed in pediatric CCA patients with congenital biliary dilatation [129].

**CEUS:** Although the CEUS features of CCA are rarely reported, malignant FLLs have similar CEUS characteristics with rapid hyperenhancement in the arterial phase followed by early and marked washout during the portal venous phase [60]. In adolescents and young adults, rim-like hyperenhancement during the arterial phase can be observed in CCA lesions due to large fibrous tissues and necrosis in the center of the lesions [130].

**CT:** CCA is usually hypodense on unenhanced CT images and shows variable enhancement on CECT and mostly shows only mild enhancement on delayed images [32]. Stone formation in the bile ducts can be revealed [129]. Chest CT is used for the detection of lung and lymph nodes metastases.

**MRI:** On unenhanced MRI, the lesions are usually slightly hypointense on T1-weighted images and slightly hyperintense on T2-weighted images, and present similar enhancement patterns to CT. The peripheral lesions show hepatic capsular retraction. Chavhan et al. [32] presented the case of a 16-year-old boy with combined CCA and HCC. With primary sclerosing cholangitis, this patient demonstrated mildly hyperintense, heterogeneous soft tissue at the portal vein extending along central portal tracts on coronal and axial T2-weighted MRI. On enhanced axial T1-weighted MRI, the lesion showed only minimal enhancement compared to the adjacent parenchyma.

**Other imaging techniques:** In advanced stages of CCA with metastatic lesions, a PET/CT scan from the skull to thigh may show numerous hypermetabolic and nonmetabolic pulmonary lesions as well as multiple hypermetabolic hepatic lesions [131].

### 5.3. Management and Prognosis

Chances of survival are related to the resectablity of the lesions, therefore, the early detection of CCA is crucial for survival as it allows for a prompt surgical approach. The surveillance of children and adolescents with biliary diseases, particularly primary sclerosing cholangitis and congenital biliary dilation, may contribute to the earlier detection of CCA and therefore potentially lead to better survival outcomes. However, patients with metastatic lesions have poor outcomes regardless of surgical intervention [116]. Overall, the 3-year overall survival of pediatric CCA was reported to be 35–50% with a median survival time of 10 months [117].

## 6. Conclusions

Malignant FLLs are commonly diagnosed in adults but should also be paid attention to in pediatric patients. Accurate descriptions of radiological and histopathological characteristics may be useful in distinguishing these lesions more precisely. CEUS has proved to be a non-invasive, safe, and effective method to locate, characterize, and make clinical follow-ups of malignant FLLs in pediatric patients.

## Figures and Tables

**Figure 1 diagnostics-13-03568-f001:**
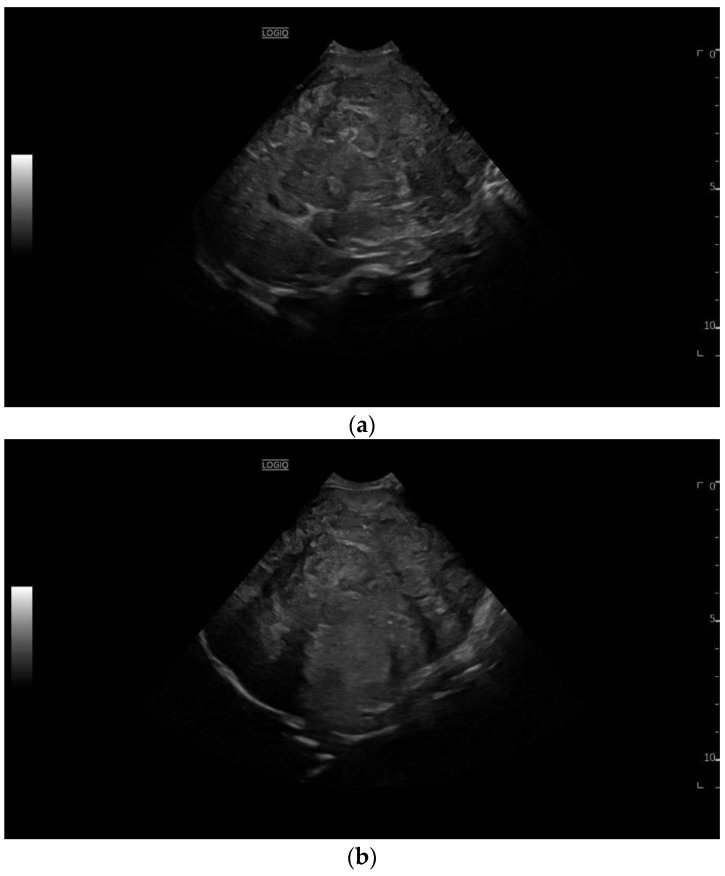
A 10-month-old male former preterm infant (24 weeks) hospitalized for a palpable upper abdominal mass. In addition to the palpable upper abdominal tumor, there was refusal to feed with dystrophy and increased sweating, and an elevated AFP level in the laboratory. Sonography revealed a large, inhomogeneous liver tumor (**a**) with partial displacement and partial infiltrative growth (**b**). Displacement of the hepatic veins and portal veins without evidence of tumor thrombi (**c**).

**Figure 2 diagnostics-13-03568-f002:**
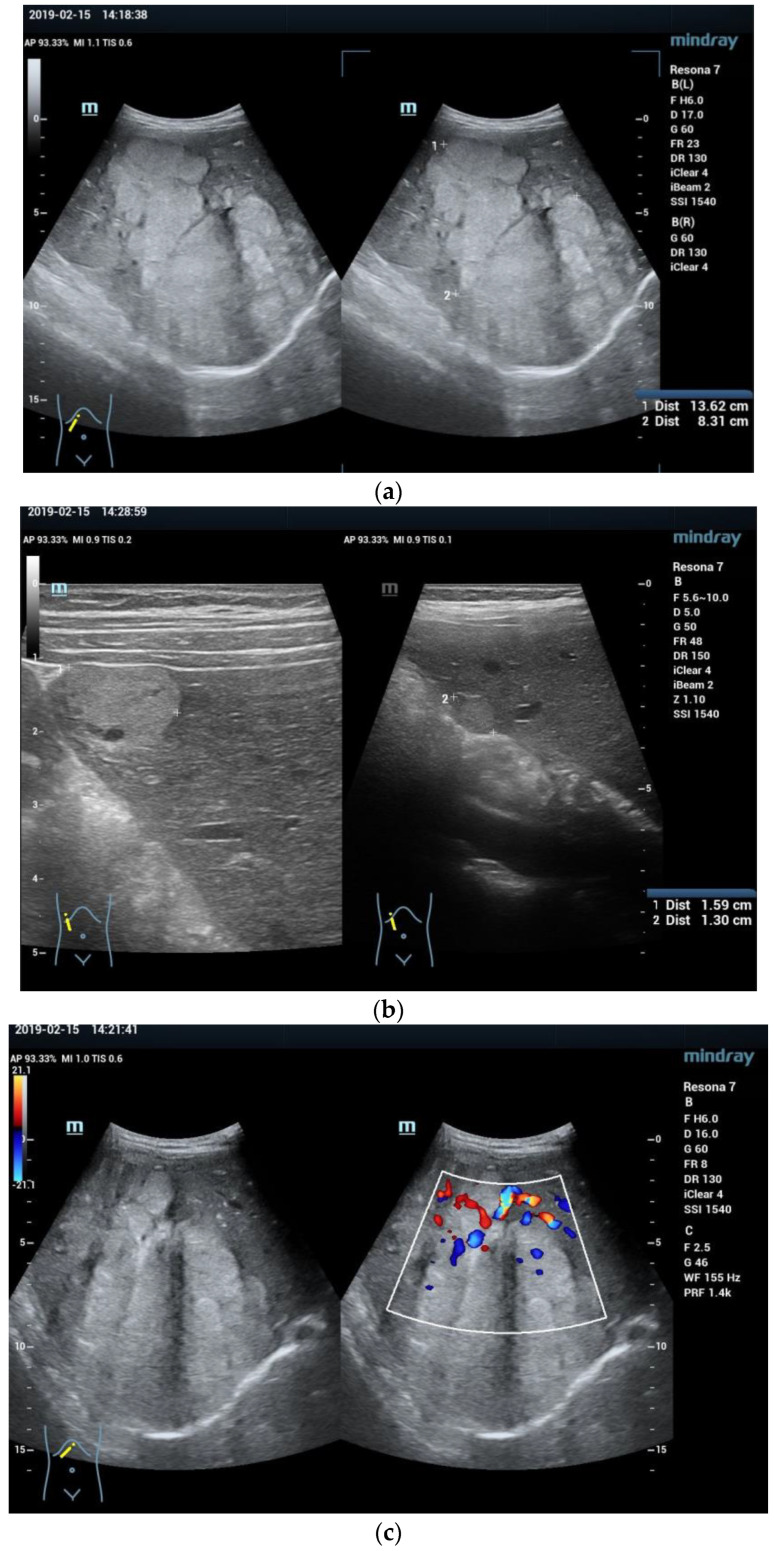
A 12 year old boy with a large liver mass, detected in the regional hospital on a scheduled ultrasound exam performed due to abdominal pain and fatigue. B mode ultrasound showed a very big 136 × 83 mm irregular mass in the right lobe (**a**) and separate nodules in segments 5 (15 mm) and 6 (13 mm) (**b**). Color flow Doppler revealed highly vascular and avascular parts within the mass (**c**). Contrast enhanced ultrasound showed chaotic hyperenhancement in the early arterial phase (**d**), with the start of wash out at the end of the arterial phase and small central necrosis (**e**). The portal venous phase revealed the progression of wash out (**f**) to prominent in the late phase (**g**). The second dose of SonoVue was administered to examine the nodule in segment 5. Rapid arterial hyperenhancement was noticed (**h**). This was followed by slight wash out in the portal venous phase (**i**) which was more expressed in the late phase (**j**). Multifocal liver malignancy was suspected. Percutaneous ultrasound guided biopsy confirmed fibrolamellar hepatocellular carcinoma (HCC) (SC5-1U transducer).

## Data Availability

Not applicable.

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
