# Peer review of "Review on Pediatric Malignant Focal Liver Lesions with Imaging Evaluation: Part I"

_diagnostics, 2023, doi:10.3390/diagnostics13233568_

Round 1

Reviewer 1 Report

Comments and Suggestions for Authors

The article discusses the importance of paying attention to malignant FLLs in pediatric patients, as they are commonly diagnosed in adults but can also occur in children. Accurate descriptions of radiological and histopathological characteristics can help distinguish these lesions more precisely. The article suggests that CEUS is a non-invasive, safe, and effective method for locating, characterizing, and making clinical follow-ups of malignant FLLs in pediatric patients.

Overall, the article is concise and to the point, providing important information about diagnosing and treating malignant FLLs in pediatric patients. However, it lacks detail and provides no specific examples or case studies to support its claims. Additionally, the article does not provide any information about the authors or their credentials, which may affect its credibility.

In conclusion, while the article provides useful information about diagnosing and treating malignant FLLs in pediatric patients, it could benefit from more detail and supporting evidence. Information about the authors and their credentials would also increase the article's credibility.

The article has some shortcomings that could affect its credibility and usefulness. These include:

1. Lack of detail: The article does not provide specific examples or case studies to support its claims. This could make it difficult for readers to understand the practical applications of the information presented.

2. Limited scope: The article focuses primarily on using CEUS to diagnose and treat malignant FLLs in pediatric patients. While this topic is important, the article could benefit from a broader discussion of other diagnostic and treatment options.

3. Incomplete references: The article does not provide complete references for all cited sources. This could make it difficult for readers to locate the original sources and evaluate the accuracy of the information presented.

Overall, while the article provides useful information about diagnosing and treating malignant FLLs in pediatric patients, it could benefit from more detail, broader scope, and complete references.

Comments on the Quality of English Language

minor

Reviewer 2 Report

Comments and Suggestions for Authors

In this interesting review, the authors aimed to address pediatric malignant focal liver lesions (FLLs) with an emphasis on imaging characteristics and the benefits of each imaging modality on the background of current knowledge and treatment strategies.

The review is well-written and presented. However, in my opinion, a review addressing the imaging evaluation of FFL, the current role of MRI in difficult cases should be further discussed. 

The authors properly discuss the addedd valuae of gadolinium ethoxybenzyl diethlenetriamine pentaacetic acid (Gd-EOB-DTPA)-enhanced MRI with hepatobiliary phase as most HCC lesions present hypointense on the hepatobiliary phase, but LI-RADS major criteria used for diagnosis of adult HCC at risk are seen in only a subset of pediatric HCC. However, other that LI-RADS criteria, the authors should also recall two well-described MR features of diagnostic value: 1) it has been demonstrated that small and hypovascular HCCs are characterized by double hypointensity in the portal/venous and hepatobiliary phases that could be considered a magnetic resonance pattern highly suggestive of hypovascular hepatocellular carcinoma; 2) Abbreviated MRI (AMRI), corresponding to the acquisition of a limited number of sequences including T1-weighted, T2-weighted, and DWI sequences, with or without contrast administration, as been suggested by recent studies as accurate screening imaging tool, as recently described (Screening of liver cancer with abbreviated MRI. Hepatology. 2023 Aug 1;78(2):670-686. doi: 10.1097/HEP.0000000000000339.).
